# Does environmental heterogeneity explain β diversity of estuarine fish assemblages? Example from a tropical estuary under the influence of a semiarid climate, Brazil

**Caroline Stefani da Silva Lima**, **Emanuelle Bezerra Maciel, Fernando José König Clark, André Luiz Machado Pessanha***

Programa de Pós-Graduação em Ecologia e Conservação, Laboratório de Ecologia de Peixes, Universidade Estadual da Paraíba, Avenida das Baraúnas, 351, Bairro Universitário, Campina Grande, PB, Brazil

* andrepessanhauepb@gmail.com

**Data Availability Statement:** All relevant data are within the paper and its Supporting Information files.

## Abstract

Estuarine fish assemblages are often sensitive to environmental conditions, because fluctuation in physico-chemical conditions at different spatial and seasonal scales can directly influence species distributions. In this way, we conducted a field survey to investigate the role of estuarine gradient (environmental heterogeneity) in fish α and β diversity. The study was carried out in three zones in Mamanguape River estuary according to salinity and geomorphology features during an atypical climatic event in 2015. In total, 18,084 specimens of 125 species were captured. Additive partitioning of diversity analysis detected a higher proportion of beta diversity among estuarine zones during the rainy ($\beta_3$ = 58.6%) and dry season ($\beta_3$ = 40.94%) and were higher than expected by chance ($Prop_{exp> obs}$ <0.001). Decomposing β-diversity analysis showed that total β-diversity ($\beta_{sor}$) results were more dominated by species turnover ($\beta_{sim}$) than nestedness ($\beta_{nes}$) in both seasons. Forward selection procedure and db-RDA identified salinity, coarse sand and chlorophyll-*a* as the main environmental variables influencing $\beta_{sor}$ and site distance from estuary mouth and split as the main landscape variables. Variation partitioning analysis revealed more contribution to the pure fraction of environmental variables to fish species turnover, however, both pure fraction of environmental and landscape variables significantly contributed to $\beta_{sim}$. Our study highlighted the importance to environmental heterogeneity and connectivity to promote fish diversity across the Mamanguape River estuary. Thus, future conservation policies should focus on maintaining these two components to guarantee its nursery ground role to estuarine fish assemblages.

## Introduction

Estuarine fish assemblages are often sensitive to environmental conditions, because fluctuations in physico-chemical conditions at different spatial and seasonal scales can have a direct influence on distribution, diversity, breeding, abundance, growth, survival, and behaviour of

**Funding:** The authors received no specific funding for this work.

**Competing interests:** The authors have declared that no competing interests exist.

fishes [1–3]. Therefore, changes in salinity gradient, turbidity, and food availability have been considered some of the main variables driving juvenile fish assemblages in estuaries [4–7], associated with rainfall and freshwater input which modify the dynamics in these ecosystems. Estuaries which are under the influence of a semiarid climate, are characterized by intermittent riverine ecosystems that cross dry lands, which alters fresh water flows and promotes the extended intrusion of seawater inside the estuarine system [8]. This is particularly common on the Brazilian northeastern coast, where the rivers originate in the semi-arid interior before reaching the tropical zone nearer the coast, and these areas receive most of their flow [9]. The effects of such changes could be responsible for shifts in fish community structure, in particular an increase in marine juvenile abundance, through what was call the hypothesis of "marinization" [10]. This issue is important because alteration of river flow regimes has potential for inducing changes in recruitment variability in marine fish during the larval and juvenile phases [11, 12], mainly for species utilizing those habitats as nursery areas [11–13] and migratory corridors [11, 14]. Additionally, decreasing river discharge has had a relatively high effect on connectivity with adjacent coastal habitats and marine fish recruitment to estuaries [13, 15], largely because ecosystem connectivity, habitat quality and environmental factors are important processes in defining fish diversity in estuaries.

Spatial patterns in diversity are driven by a hierarchy of factors operating at multiple scales [16]. Spatial environmental heterogeneity influences species coexistence at small scales based on the niche differentiation concept, generally associated with increased resource availability, provision of refuges and changes in environmental conditions, which in turn should promote differences in species composition and abundance [17]. Thus, these mechanisms promote increased local diversity levels [17, 18].

Additive partitioning of diversity has been suggested as an alternative method for quantifying beta diversity at various spatial scales using null models (random distribution of species) to discover diversity dependence on ecological processes in niches or neutrals that are important for generating biodiversity patterns [19–21]. This method aims to reveal the contribution of alpha diversity and beta diversity in shaping diversity [22]. However, while alpha diversity describes only the local diversity, it does not account for a variation in species composition between sites, which could indicate the underlying processes on diversity across gradients [23, 24]. Therefore, beta diversity has been used to quantify dissimilarity of species among multiple-sites [25], and to understand which ecological processes drive species distribution and biodiversity [26, 27].

Beta diversity can be decomposed into two components: nestedness and turnover [28–30]. Nestedness occurs when species that are present on the least diverse site are a subset of the most diverse sites [31], which might be defined by stochastic processes creating species exclusion in the habitat (extinction, dispersion ability and migration) [32]. Turnover occurs due to species substitution in spatial or environmental gradients mainly influenced by habitat filtering [25, 33, 34]. Therefore, shedding light on the contribution of these antithetical phenomena that generate beta diversity is fundamental in understanding how communities respond to environmental variations and spatial and seasonal changes [35].

Several studies have assessed which factors influencing the partitioning of diversity at multiple scales have been found to have important implications for potential applications in conservation biology [36–38]. These studies are mainly concerned to identify priority areas within target regions for strategies for conservation policies, including, habitat quality, habitat connections (corridors and stepping stones), and spatial heterogeneity [39–41]. Additionally, when turnover or nestedness are observed in ecosystems, they are valuable targets in themselves and can contribute particularly to the process that drives these patterns to ensure species diversity [38].

We conducted a field survey to investigate the role of estuarine gradient (environmental heterogeneity) on fish α and β diversity in a tropical estuary in Brazil. Many recent studies using this approach concluded that estuarine fish assemblages respond strongly to environmental heterogeneity, demonstrating that a high dissimilarity in species composition and species dominance occurs both at regional and local scales [42–44]. Thus, the aim of this paper was to evaluate beta diversity patterns of fish assemblages over space (estuarine gradient) and time (seasons) and to investigate the relationship between environmental and spatial components in determining spatial patterns of variations in beta diversity in a tropical estuary. We hypothesized, that during rainy season would increase beta diversity because of species substitution along the estuarine gradient is expected to promote turnover among assemblages, e.g., through the replacement of some species by others as a consequence of environmental variables.

## Material and methods

### Study area

The Mamanguape River estuary is located on the north coast of Paraíba State, Brazil. The estuarine area is 25 km in length, including the Environmental Protection Area of the Mamanguape River (EPA Decree 924/1993), with an area of 146.4 km$^2$ [45]. The estuarine channels are covered by mangrove forests that include the species *Rhizophora mangle*, *Avicennia germinans*, *Avicennia schaueriana*, *Laguncularia racemosa*, and *Conocarpus erectus* [46]. In the estuary entrance, there is a long perpendicular reef barrier to the shoreline, creating a semiclosed bay with calm waters and a mix of freshwater and seawater.

The aim of the EPA creation was to protect marine manatees, which use the estuary as the main breeding site in the Northeast Region of Brazil [47], as well as to create a natural resource conservation area [48]. However, this estuary shows anthropogenic impacts caused by sugarcane cultivation [49], shrimp farming [50] and mangrove deforestation [49, 51]. Recently, microplastics were reported in the stomach contents of several fish species in this estuary [52].

The climate of the region is hot and humid (AS' de Köppen) [53], with the rainy season between February and July (average precipitation is 161.4 mm; data from 2010 to 2016) and the drought between August and January (average precipitation is 57.9 mm; data from 2010 to 2016) [54]. The normal rainfall is between 1750 and 2000 mm annually, and the average temperature is 24–26˚C [53]. In this estuary, the salinity varies between 2 to 35, from the upper to lower region, respectively [55]. The tides in this region are characterized by semi-diurnal tides and mesotidal regimes, with the mean spring tidal range from 0.2 m (low water) to 2.2 m (high water) [56].

### Fish sampling

The sampling survey consisted of five excursions carried out during three rainy-season months (May, June and July 2015) and three rainy-season months (October and November 2015 and January 2016). All samples always occurred during spring low tide. Samplings were performed in different zones of estuary based on general characteristics of tidal channel and salinity sectors: **Zone 1** was located in the upstream section of the estuary, and is characterized by contains a narrow and shallow channel and salinity between 3 and 5, and generally mangrove forest structure form narrow fringe communities along the shores; **Zone 2** also was in the upstream section of the estuary and the channel dimensions are broader and with shallow depth, salinity between 17 and 27, and the channel is bordered by extensive mangrove forest; and **Zone 3** was located in downstream, characterized by deeper water and an influx of high

salinity water (between 30 and 37), and most of the zone contain seagrass beds, numerous, and sometimes extensive mud and sand flats nearest to the estuary mouth.

The fish were collected with a beach seine net (10 m long, 1.5 m high and with 0.8 cm mesh) and the hauls were taken parallel to the coastline at a depth of approximately 1.5 m. We performed nine samples in each of the three zones. The sample design contained a total of 162 samples (3 zones x 3 points x 3 sampling sites x 6 months). Captured fishes were anaesthetized in a solution 4.5 mg/mL of lidocaine hydrochloride and then kept in a cooler with ice. Thereafter, fish were fixed in formalin solution 10% and after 48 h, transferred to ethanol 70% solution.

## Ethics statement

Animal care and handling were carried out following institutional guidelines according to Brazilian laws by SISBIO Collection of Species Permit number 24557-27/10/2010 issued by ICMBio, Brazilian Environmental Agency, and follows the ethics rules applicable to the of animals in teaching and for research based on the of Brazilian law (Federal Law 11.794 of October 08, 2008)

## Environmental variables

At each sampling site, we recorded salinity, water temperature (˚C), pH, dissolved oxygen (DO, mg/L), in situ using a multiparameter sensor. Transparency (cm) and depth (cm) were measured using a Secchi disc and an Echotest depth sounder, respectively.

The substrate of the sites was classified in four categories based on grain size analysis: very coarse sand (VCS: > 2.0 mm), coarse sand (CS: 0.5 to 2.0 mm), medium sand (MS: 0.25 to 0.5 mm), and fine sand (FS: <0.125 mm) [57]. Percentage organic matter content was quantified through the difference between the weight before and after burning of 3 g sediment samples by 500˚ C during 8 h in a moufle. In addition, chlorophyll-*a* was extracted by acetone method, and then, its concentration was measured spectrophotometrically [57]. Precipitation data were compiled from the Executive Agency for Water Management of the State of Paraiba (AESA website: www.aesa.pb.gov.br).

## Landscape variables

To characterize the landscape variables of the estuary, we measured canal width and site distance from the estuary mouth through Google Earth Pro software. Width was measured by distance between banks at each sampling site and proximity by distance between sampling site position to estuary mouth. In both landscape variables, at several sites, two or three stations were measured. In addition, meanders were calculated through the ratio of the number of bends (the directional changes observed along the main channel) to the length of the main channel and divided by the ratio of the number of branches to the length of the main channel [58]. We also extract variable elevation data for each zone from the results of Medeiros [59] and variables of agriculture, aquaculture, urban, mangrove, restinga, rainforest and waterbody data were extracted from the results of Teixeira et al. [60].

## Data analysis

**Environmental and landscape variables.** Permutational Multivariate Analysis (PERMANOVA) was used to investigate whether environmental variables differed among zones and seasons. The environmental data were log (x + 1) transformed or underwent arcsine transformation (only for the percentage data) to reduce the influence of extreme values on the analysis.

The PERMANOVA included two fixed factors: zones with three levels and seasons with two levels. PERMANOVA tests were performed with 9,999 permutations to ensure the significance level [61]. We performed these methods using functions from the "vegan" packages in the R statistical environment [62, 63].

**Alpha diversity.** Rarefaction and extrapolation curves were built for fish assemblages for zones in each season in order to determine the accumulation of species with sample size. Non-parametric estimators have been useful to standardize the number of samples and enabled prediction of the real diversity considering the expected number of species undetected by the sampling effort [64]. For these analyses, we consider for individual-based data the number of individuals. As suggested by Chao et al. [65], extrapolation sample size was double the reference sample size for each zone during seasons. Species diversity was based on richness species ($q = 0$) with 100 bootstrap replications and 95% confidence intervals. Significances of spatial and seasonal variations for species richness were also tested by univariate PERMANOVA based on euclidean distance matrix utilizing the same design considered for abiotic variables. We performed these methods using functions from "iNEXT" [66], and "vegan" packages through the R statistical environment [63].

**Additive partitioning of diversity.** Total fish diversity partitioned within its components and extended through the sampling design's spatial scales: sampling sites, points and zones. In this case, a nested hierarchical sampling design was established for the study: $\alpha$ diversity corresponds to the diversity within the sampling site, $\beta_1$ corresponds to the variation of diversity between the sample subunits, $\beta_2$ and $\beta_3$ correspond to the variation of diversity between points and between estuarine zones, respectively. The regional diversity of the estuary during the rainy and dry seasons was determined by summing the components of diversity $\alpha$ and $\beta$ ($\gamma = \alpha + \beta_1 + \beta_2 + \beta_3$) [20]. Thus, it was possible to obtain the beta diversity proportions in the same unit at each spatial level considered and could be directly compared.

An individual-based null model type was used to determine if there were differences between the results of the randomly observed and expected components using 9,999 randomizations [20]. In this case, when the randomly obtained values are higher than the observed values ($Prop_{exp>obs} > 0.975$) this indicates that the observed values were significantly lower than the randomly expected ones. On the other hand, when the randomly obtained values were lower than those observed ($Prop_{exp>obs} < 0.025$), this indicates that the observed values were significantly higher than the randomly expected ones. We performed additive partitioning of diversity analysis through statistical software Partition 3.0 [67].

**Components of beta diversity: Turnover and nestedness.** Sorensen dissimilarity index family was computed from the presence/absence matrix to identify whether beta diversity ($\beta_{sor}$) among zones was mainly generated by turnover ($\beta_{sim}$) or nestedness ($\beta_{nes}$) [26]. Sorensen dissimilarity index was calculated for the rainy and dry season, separately, to observe differences in spatial pattern. The value of $\beta_{sor}$ and its components range from 0 to 1, indicating total similarity or total dissimilarity, respectively [30]. The sum of the values of the $\beta_{sim}$ and $\beta_{nes}$ components is equal to the value of the $\beta_{sor}$ obtained in each test performed [30]. These analyses were performed utilizing functions of packages "vegan" [62] and "betapart" [26] through the R statistical environment [63].

**Drivers of beta diversity components.** To find which subset contributes more to the pattern of beta diversity, variables were separated into two subsets: environmental and landscape variables. *A priori*, variables data were tested to significance through distance-based redundancy analysis (db-RDA) to association with beta diversity pairwise dissimilarity matrices based on presence/absence data ($\beta sor$) [26]. Forward selection procedure was performed to select a model excluding variables with adjusted $R^2$ greater than global model and *p* value greater than 0.05 with 9,999 permutations [68]. Multicollinearity among environmental and

landscape variables separately was tested using the variance inflation factor (VIF), and those with VIF > 10 were omitted from the analysis [69]. Afterwards, variation partitioning analysis was performed to evaluate contributions of pure environment variables subsets fractions (E|S), pure environment variables subsets fractions (S|E), shared effects (E+S) and the unexplained variance (U) on beta diversity components [70, 71].

All analyses above were performed utilizing functions of packages "vegan" [62] and "beta-part" [26] through the R statistical environment [63].

## Results

### Environmental variables

The environmental and landscape variables showed a distinct spatial and seasonal variation (Table 1). Some environmental variables have strong spatial structure along the river-ocean gradient, as such salinity, transparency, dissolved oxygen, and organic matter were consistently higher in zone 3 than those registered in zone 1. This gradient is consistently more marked during the dry season. Depth, very coarse sand, fine sand, and width also had higher values in zone 3, while coarse sand had a higher value in zone 1 and medium sand in zone 2. During the rainy season, depth, medium sand, fine sand, width and site distance from estuary mouth had

**Table 1. Mean variations in the environmental and landscape variables (±SE) for the study period in the Mamanguape River estuary.** (Z1 = zone 1; Z2 = zone 2; Z3 = zone 3).

| Variables | Rainy | | | Dry | | |
|---|---|---|---|---|---|---|
| | **Z1** | **Z2** | **Z3** | **Z1** | **Z2** | **Z3** |
| **Environmental** | | | | | | |
| Temperature (˚C) | 29.00 ± 0.43 | 27.54 ± 0.3 | 28.16 ± 0.42 | 30.29 ± 0.3 | 28.74 ± 0.26 | 29.30 ± 0.43 |
| Salinity | 3.18 ± 0.83 | 5.74 ± 1.38 | 27.43 ± 0.42 | 5.20 ± 0.30 | 17.53 ± 1.48 | 39.15 ± 1.21 |
| Transparency (cm) | 27.22 ± 3.93 | 21.74 ± 2.52 | 40.74 ± 1.40 | 42.78 ± 0.68 | 46.67 ± 2.97 | 55.19 ± 4.76 |
| pH | 8.21 ± 0.22 | 8.96 ± 0.12 | 8.44 ± 5.54 | 9.89 ± 0.30 | 9.24 ± 0.21 | 10.74 ± 0.89 |
| Dissolved oxygen (mg/L) | 5.29 ± 0.22 | 5.14 ± 0.19 | 6.26 ± 0.11 | 20.73 ± 5.21 | 10.95 ± 3.14 | 26.40 ± 7.40 |
| Organic matter (%) | 15.21 ± 2.85 | 11.40 ± 3.27 | 25.10 ± 0.31 | 3.24 ± 0.33 | 7.33 ± 1.46 | 20.82 ± 2.48 |
| Chlorophyll-*a* (μg/L) | 3.88 ± 0.42 | 7.79 ± 1.52 | 4.11 ± 4.38 | 26.74 ± 9.00 | 29.44 ± 9.58 | 28.41 ± 9.62 |
| Depth (cm) | 58.52 ± 4.85 | 52.85 ± 4.80 | 69.26 ± 3.81 | 46.11 ± 3.55 | 57.41 ± 4.91 | 73.70 ± 5.18 |
| Very coarse sand (%) | 5.48 ± 0.63 | 4.95 ± 0.85 | 13.31 ± 4.16 | 5.72 ± 0.61 | 8.90 ± 2.88 | 17.17 ± 2.20 |
| Coarse sand (%) | 34.17 ± 3.03 | 31.36 ± 3.28 | 23.09 ±3.25 | 57.51 ± 3.06 | 46.47 ± 3.69 | 33.03 ± 4.03 |
| Medium sand (%) | 33.71 ± 2.10 | 44.50 ± 2.14 | 32.61 ± 2.25 | 28.52 ± 2.23 | 35.68 ± 3.18 | 27.94 ± 3.03 |
| Fine sand (%) | 46.33 ± 4.78 | 31.29 ± 3.19 | 45.62 ± 3.26 | 13.67 ± 1.78 | 15.58 ± 2.62 | 38.76 ± 5.68 |
| **Landscape** | | | | | | |
| Elevation (m) | 11 | 11 | 6 | 11 | 11 | 6 |
| Agriculture (m$^2$) | 832.33 | 832.33 | 759.95 | 832.33 | 832.33 | 759.95 |
| Aquaculture (m$^2$) | - | - | 120.63 | - | - | 120.63 |
| Urban area (m$^2$) | 72.38 | 72.38 | 108.56 | 72.38 | 72.38 | 108.56 |
| Mangrove (m$^2$) | 989.14 | 989.14 | 1411.34 | 989.14 | 989.14 | 1411.34 |
| Restinga (m$^2$) | - | - | 627.26 | - | - | 627.26 |
| Rainforest (m$^2$) | 48.25 | 48.25 | 0.00 | 48.25 | 48.25 | - |
| Waterbody (m$^2$) | 132.69 | 132.69 | 446.32 | 132.69 | 132.69 | 446.32 |
| Distance from estuary mouth (m) | 23657.7±122.1 | 18380.6±110.35 | 2640.04±183.3 | 22037.04±131.5 | 19049.61±158.20 | 2571.8±160.8 |
| Canal width (m) | 34.13 ± 1.16 | 264.58 ± 11.79 | 2,798.67 ± 322.4 | 44.54 ± 2.67 | 195.83 ± 11.44 | 1,784.64 ±267.45 |
| Meanders (x10$^{-3}$) | 1.09 | 8.04 | 4.66 | 1.09 | 8.04 | 4.66 |
| Split (x10$^{-3}$) | 1.09 | 1.61 | 4.66 | 1.09 | 1.61 | 4.66 |

higher values, but very coarse sand and coarse sand had higher values in the dry season (Table 1). Temperature and pH were usually the same in all zones and periods. PERMANOVA test detected significant spatial differences only salinity (F = 125.32; $p$ = 0.0001), transparency (F = 7.94; $p$ = 0.0003), dissolved oxygen (F = 3.48; $p$ = 0.03), organic matter (F = 22.72; $p$ = 0.0001), depth (F = 8.18; $p$ = 0.0003), very coarse sand (F = 13.01; $p$ = 0.0001), coarse sand (F = 15.00; $p$ = 0.0001), medium sand (F = 8.1; $p$ = 0.0006), fine sand (F = 11.50; $p$ = 0.0001). Significant seasonal differences were observed in temperature (F = 38.43; $p$ = 0.0001), salinity (F = 50.71; $p$ = 0.0001), transparency (F = 44.11; $p$ = 0.0001), pH (F = 17.89; $p$ = 0.0001), dissolved oxygen (F = 25.1107; $p$ = 0.0001), organic matter (F = 10.1; $p$ = 0.001), chlorophyll-$a$ (F. = 22.54; $p$ = 0.0001), coarse sand (F = 36.45; $p$ = 0.0001), medium sand (F = 7.95; $p$ = 0.007) and fine (F = 32.27; $p$ = 0.0001).

For the landscape variables, elevation, distance, agriculture, and rainforest were more pronounced at the upper estuary (considering zones 1 and 2) than zone 3. Aquaculture, urban, mangrove, restinga, waterbody, canal width and split had higher values in zone 3. Only meander had a higher value in zone 2. Only significant spatial differences were detected in all landscape variables: canal width (F = 546.27; $p$ = 0.0001), distance (F = 2326.43; $p$ = 0.0001), split (F = 1.21; $p$ = 0.0001), meanders (F = 4.28; $p$ = 0.0001), elevation (F = 5.42; $p$ = 0.0001), agriculture (F = 5.72; $p$ = 0.0001), aquaculture (F = 5.81; $p$ = 0.0001), urban (F = 5.56; $p$ = 0.0001), mangrove (F = 5.72; $p$ = 0.0001), restinga (F = 5.51; $p$ = 0.0001), rainforest (F = 5. 60; $p$ = 0.0001) and waterbody (F = 5.26; $p$ = 0.0001).

## Alpha diversities

A total of 18,084 fish from 125 species were captured in a total of 162 seine hauls (See S1 Appendix). *Atherinella brasiliensis* (26.91%), Engraulidae larvae (19.64%) and Gerreidae larvae (7.43%) dominated the catches numerically, and together accounting for 53.98% of the total catch. There was a distinct spatial and seasonal trend in abundance and species richness. Most fishes were collected in zone 3 (n = 7,983), followed by zone 2 (n = 7,161) and zone 1 (n = 2,940). Fish species richness was highest in the dry season, and highest values occurred in zone 3 (n = 125), followed by zone 2 (n = 37) and zone 1 (n = 34). The species rarefaction curve showed no apparent asymptote, but a trend towards stabilization in all zones in both seasons (Fig 1). PERMANOVA found only significant differences for species richness among zones (F = 29.8; $p$ = 0.0001).

Spatial differentiation in terms of species abundance was apparent in both seasons. In rainy season for example, *Sciades herzbergii*, *Atherinella brasiliensis*, *Eucinostomus melanopterus* Engraulidae, and Gerreidae larvae were the most collected species in zone 1 contributing 55.26% of abundance, whereas in zone 2 only Engraulidae larvae and *Atherinella brasiliensis* contributed 75.88%, and Gerreidae larvae, *Atherinella brasiliensis*, *Anchoa spinifer*, and *Caranx latus* accounted for 66.1% of abundance. In the dry season, only *Atherinella brasiliensis* had a higher abundance in zones 1 and 2 (58.13% and 67.55%, respectively), whereas in zone 3 *Atherinella brasiliensis*, *Rhinosardinia bahiensis*, *Anchoa januaria*, *Harengula clupeola*, *Caranx latus*, and *Anchoa hepsetus* accounted for 54.45% of abundance.

## Additive partitioning of diversity

Additive partitions of species richness for sample units (α) showed, in general, similar results for both rainy and dry season collections (13.36 and 11.87, respectively). This suggests the presence of a large number of common species in both seasons. It was noticed that the proportions were lower than expected by chance in the null model on both seasons (Prop$_{exp>obs}$> 0.9999).

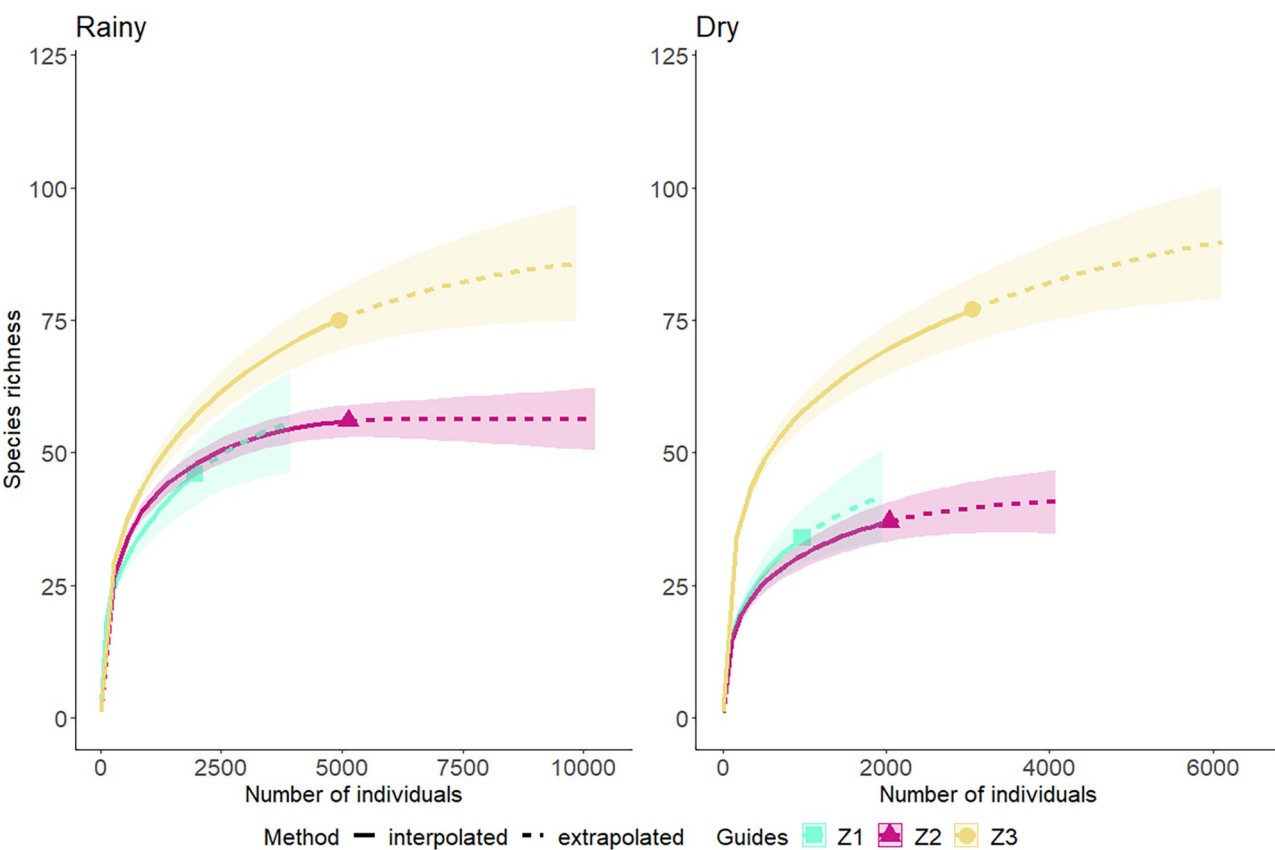

**Fig 1. Species rarefaction curves with interpolation and extrapolation of fish assemblages by three zones during 2015/2016 rainy and dry seasons in Mamanguape River estuary.**

Overall, the highest proportions of beta diversity occurred between the estuarine zones during the rainy season ($\beta_3$ = 58.6%) and the dry season ($\beta_3$ = 40.94%) followed by point sites dissimilarity and between sampling sites during the dry period ($\beta_2$ = 29.14%; $\beta_1$ = 17.01%) and rainy ($\beta_2$ = 16.98%; $\beta_1$ = 10.34%). The observed values of the $\beta_3$ component were higher than those randomly expected in both seasonal periods ($\text{Prop}_{\text{exp>obs}}$ <0.001). In the $\beta_2$ hierarchical scale, the observed and expected values were similar to each other during the rainy season ($\text{Prop}_{\text{exp>obs}}$ = 0.0001), while in the dry season the observed values were higher than randomly expected ($\text{Prop}_{\text{exp>obs}}$ <0.001). The observed proportions of dissimilarity between the sample subunits ($\beta_1$) were similar to those expected at random in the rainy season ($\text{Prop}_{\text{exp>obs}}$ = 0.9999) and dry ($\text{Prop}_{\text{exp>obs}}$ = 0.3808; Fig 2).

## Turnover and nestedness

Overall, β-diversity analysis showed a high level of dissimilarity along the estuary. Spatially, total β-diversity ($\beta_{\text{sor}}$) averaged 0.84 and 0.85 in rainy and dry seasons, respectively. Turnover ($\beta_{\text{sim}}$) was also the more predominant process of total dissimilarity than nestedness ($\beta_{\text{nes}}$) contributing to 87.68% and 12.31% in the rainy and 87.60% and 12.39% in the dry seasons, respectively (Fig 3).

## Drivers of beta diversity components

Forward selection criterion for each data subset listed four significant environmental variables (salinity, coarse sand, chlorophyll-*a*, and very coarse) and two landscape variables (distance

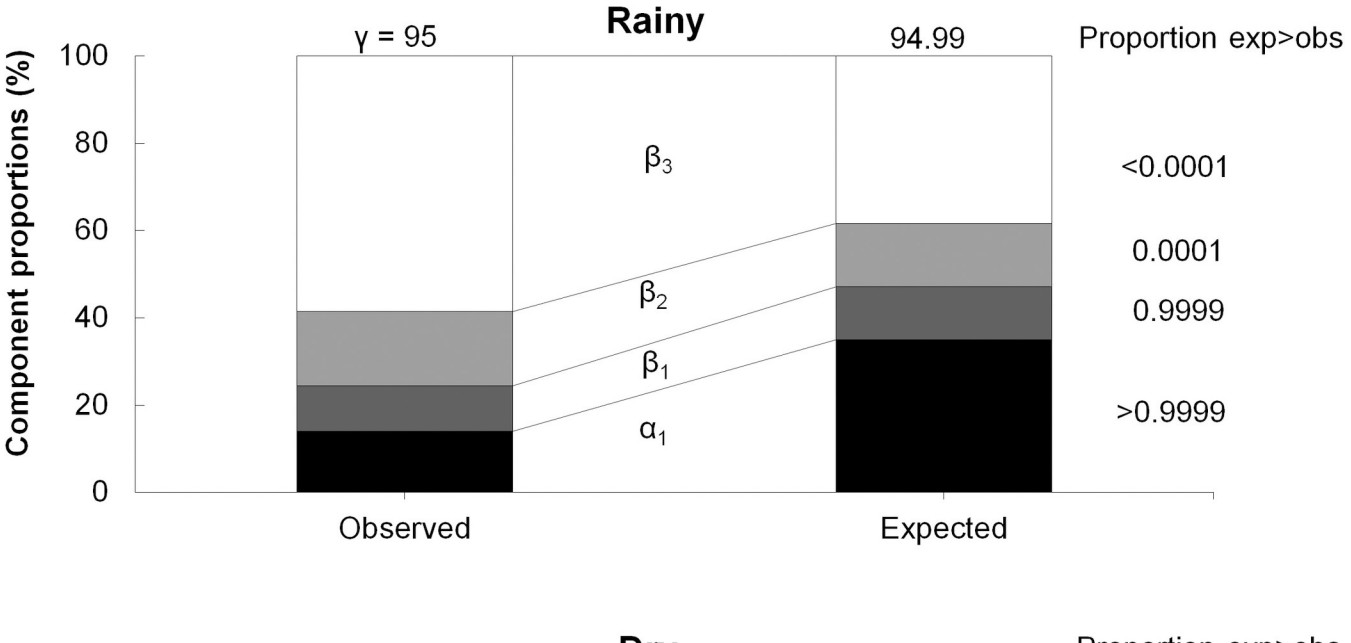

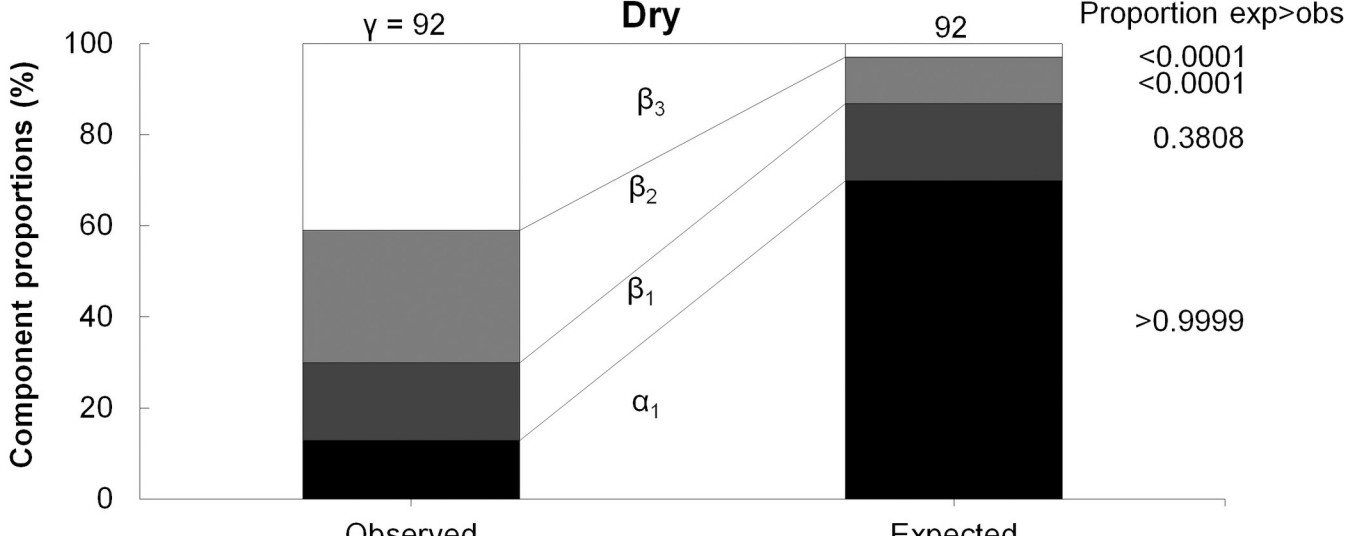

**Fig 2. γ-diversities additively partitioned into β- and α-diversities with observed and expected significance.** γ-diversity = regional diversity from estuary during rainy and dry seasons, $\beta_3$-diversity = diversity among zones, $\beta_2$-diversity = diversity among point sites, $\beta_1$-diversity = diversity among sampling sites and $\alpha_1$-diversity = diversity within sampling sites.

and split) (Table 2). The two axes of db-RDA explained 78.15% of the variation when β-diversity was tested only with environmental variables, while explaining 100% of the variation when tested only with landscape ones (Table 3).

According to db-RDA ordination, spatial separation was more evident than seasonal separation (Fig 4). Samples of zone 1 and partially zone 2 were plotted on the left side related to the occurrence of species Engraulidae larvae, *Anchoa spinifer*, *Mugil liza*, *Achirus lineatus*, *Centropomus parallelus*, and *Hyporhamphus roberti* which preferred the highest values of coarse sand and chlorophyll-*a* in these zones. The other part of the samples of zone 2 and zone 3 were plotted on the right side represented by the occurrence of species *Caranx latus*, Gerreidae larvae, *Lycengraulis grossidens*, and *Hyporhamphus unifasciatus* which were related to the coarsest

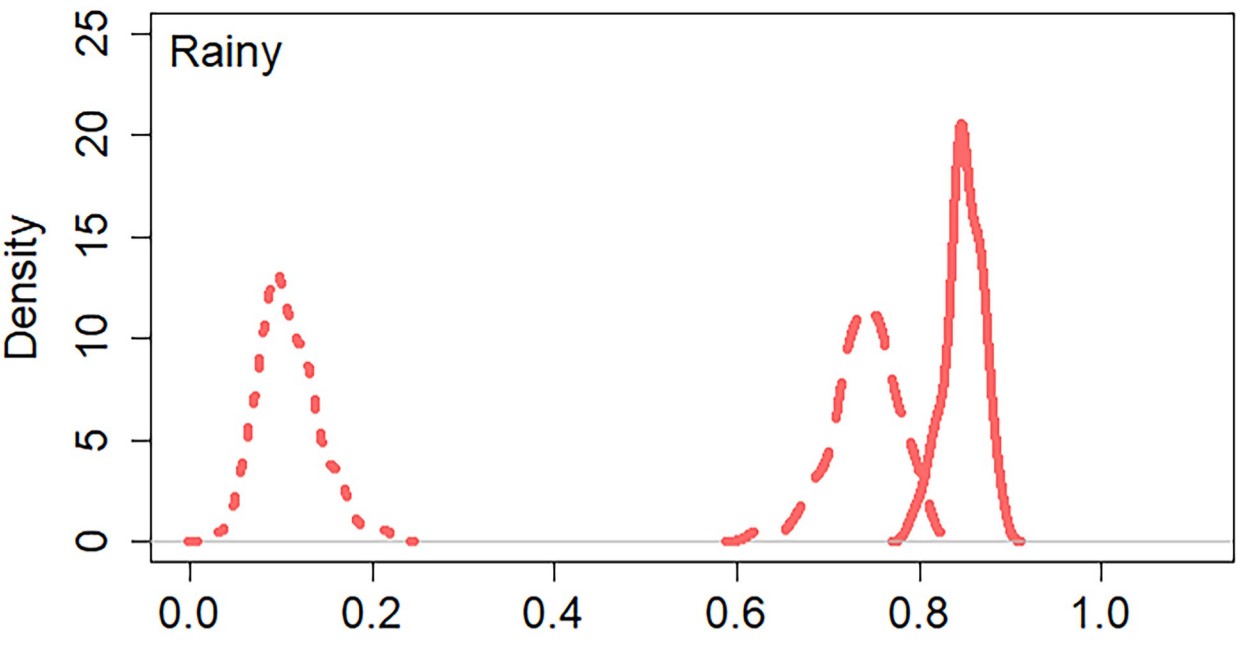

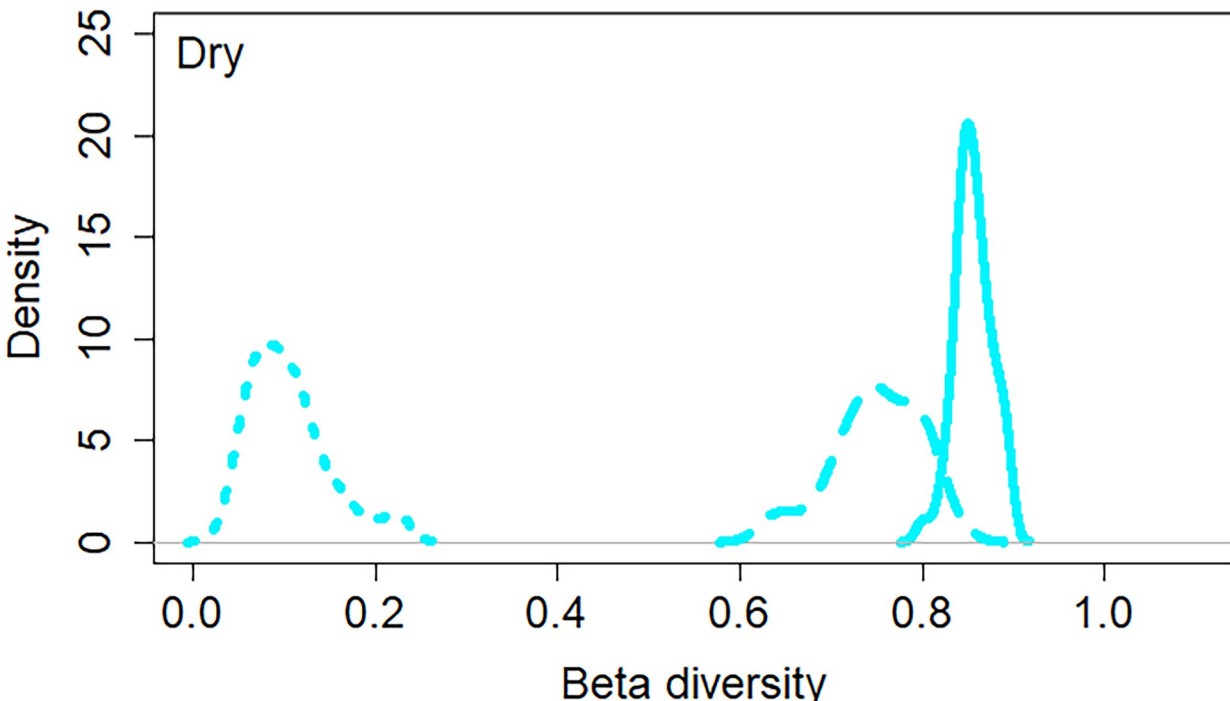

**Fig 3. Decomposed total β-diversity between turnover and nestedness components in Mamanguape River estuary during rainy (red) and dry (lightblue) seasons.** Continue lines represent total beta diversity, dashed lines turnover and dotted lines nestedness.

sand (Fig 4A). Seasonally, *A. lineatus*, *H. roberti*, *H. unifasciatus*, *C. parallelus*, *L. grossidens*, *U. lefroyi*, and Gerreidae larvae were related to the lowest values of coarse sand, chlorophyll-*a* and salinity in the rainy season which were positioned at the lower side of the plot. Samples of dry

**Table 2. Environmental and landscape variables selected by forward selection method on fish assemblage β-diversity.** Adjusted $R^2$ values indicated variation explained of each variable in each subset for db-RDA analysis.

| Variables | Variables | AIC | $R^2_{adj}$ | p-value |
|---|---|---|---|---|
| **Environmental** | Salinity | 121.26 | 0.07 | 0.0001 |
| | Coarse sand | 119.16 | 0.05 | 0.0002 |
| | Chlorophyll-*a* | 118.22 | 0.03 | 0.0004 |
| | Very coarse sand | 117.69 | 0.02 | 0.004 |
| **Landscape** | Distance from estuary mouth (m) | 117.88 | 0.13 | 0.0001 |
| | Split | 118.05 | 0.01 | 0.04 |

season were positioned at the upper side related to the occurrence of Engraulidae larvae, *A. spinifer*, *M. liza*, and *C. latus* associated to the highest values of very coarse sand (Fig 4B). Regarding landscape variables, samples of zone 1 and zone 2 were represented by the occurrence of *A. lineatus*, *H. roberti*, and *Ctenogobius boleosoma* which were related to the longest distance and lower values of split while samples of zone 3 were associated to *C. latus* and *Albula vulpes* leptocephalus larvae which were related to the shortest distance and highest values of split in the estuary. No seasonal pattern could be observed for the landscape variable in dbRDA ordination due to low variation in this data (Fig 4C and 4D).

Variation partitioning analysis showed that two subsets of variables explained 22%, 21% and 30% of variation on total beta diversity, turnover and nestedness of fish species in the estuary, respectively. Both pure environmental and landscape variables contributed significantly to beta diversity and turnover. The shared fraction of environmental and landscape variables contributed more to beta diversity and nestedness while the pure fraction of environmental variables contributed more to turnover (Fig 5).

## Discussion

Additive partitioning of fish richness evidenced that species dissimilarity in both spatial and seasonal scales led to species turnover. Spatial and environmental variables combined accounted for 17% of the variation in species turnover, and environmental variables alone accounted for much more of the variation than landscape variables. These results highlight the importance of heterogeneity and seasonal dynamics of environmental variables in establishing

**Table 3. Distance-based redundancy analysis (db-RDA) results for relationships of environmental and landscape variables with fish assemblage β-diversity in Mamanguape estuary.**

| Variables | Variables/Components | Eigenvectors coefficients | |
|---|---|---|---|
| | | Db-RDA 1 | Db-RDA 2 |
| **Environmental** | Salinity | 0.665 | -0.737 |
| | Coarse sand | 0.-685 | -0.331 |
| | Chlorophyll-*a* | -0.115 | -0.703 |
| | Very coarse sand | 0.459 | -0.035 |
| | Eigenvalues | 1.21 | 0.6 |
| | Proportion explained (%) | 52.25 | 25.90 |
| | Constrained inertia | 2.316 | |
| **Landscape** | Distance from estuary mouth | -0.985 | 0.1724 |
| | Split | -0.73 | 0.683 |
| | Eigenvalues | 1.44 | 0.24 |
| | Proportion explained (%) | 85.53 | 14.47 |
| | Constrained inertia | 1.686 | |

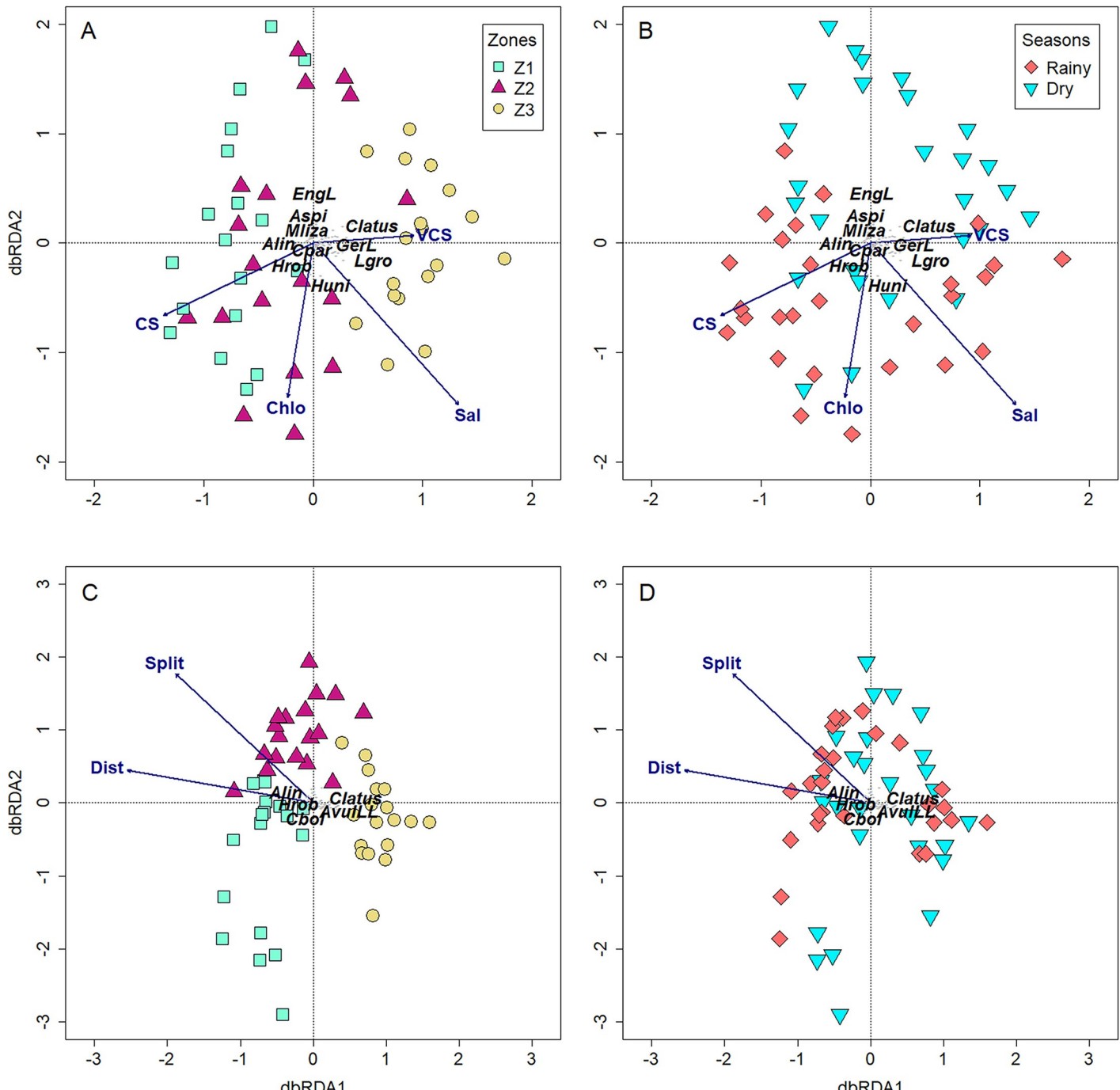

**Fig 4.** db-RDA ordinations summarizing relationships among environmental (A and B) and landscape (C and D) variables, separately, with fish assemblage β-diversity. Zones: Z1 = aquamarine, Z2 = violet and Z3 = yellow. Seasons: Rainy = red and Dry = lightblue. Variable were coded either by first 3–4 letters or by initials: Chlo = Chorophyll *a*, CS = Coarse sand, Dist = Distance from the estuary mouth, Sal = Salinity, VCS = Very coarse sand. Species were coded by the first letter of genus or species and first 3 letters of the epithets, except when the epithets were short, then, the names were completely adopted: Alin = *Achirus lineatus*, AvulLL = *Albula vulpes* leptocephali larva, Aspi = *Anchoa spinifer*, Clatus = *Caranx latus*, Cpar = *Centropomus parallelus*, Cbol = *Ctenogobius boleosoma*, EngL = Engraulidae larva, GerL = Gerreidae larva, Hrob = *Hyporhamphus roberti*, Huni = *Hyporhamphus unifasciatus*, Lgro = *Lycengraulis grossidens*, Mliza = *Mugil liza*.

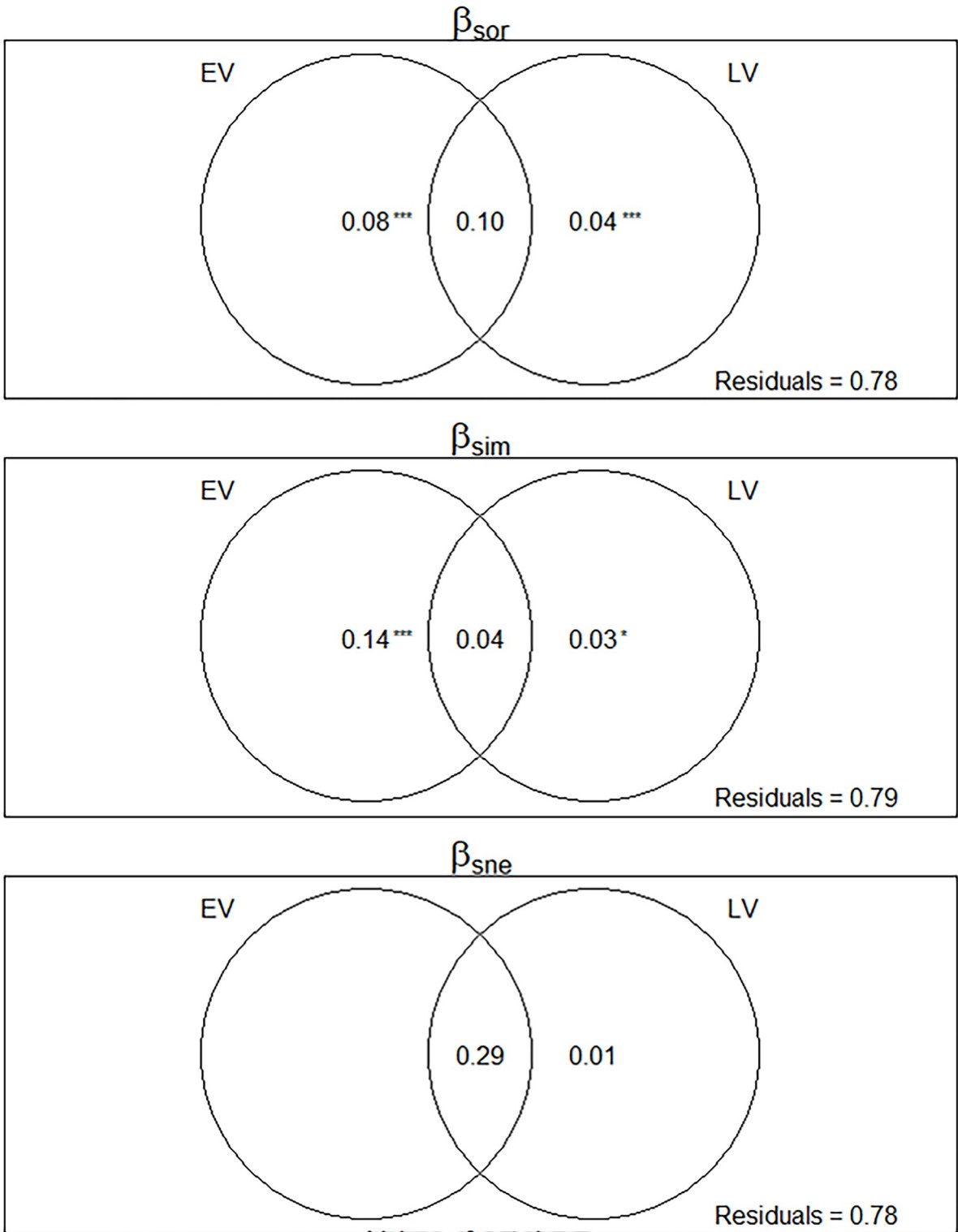

**Fig 5. Venn diagrams of variation partitioning of β_sor, β_sim and β_sne explained by environmental variables and landscape variables.**
Values in circles represent partitions of variation explained by variable subsets. Areas of overlap between circles represent shared variation between variable subsets. Variable fractions coded by initials: EV = Environmental variables (salinity, coarse sand chlorophyll-*a*, very coarse sand) and LV = Landscape variables (distance and split).

fish assemblages in this tropical estuary. Teichert et al. [43] concluded that estuarine conditions are the primary drivers of β-diversity, particularly in regards to marine connectivity and system size that promote environmental heterogeneity. Our hypothesis that fish assemblages would follow turnover among zones mostly driven by species replacement in suitable habitats across gradients was supported.

In general, fish abundance and species richness are locally associated to variability in environmental conditions, mainly salinity, primary productivity and substrate type [55, 72], which often forces a high species turnover across space in tropical estuaries. Our results of db-RDA demonstrated the influence of environmental factors on the different species of fishes in each zone (habitat conditions). In this study, resident species, such as *A. lineatus* and *E. melanopterus*, were caught in higher abundance in the upper estuary, likely a reflection of euryhaline condition. On the other hand, *C. latus*, *S. greeleyi*, *L. grossidens* were caught more consistently in zone 3 (lower estuary), highlighting the importance of the stenohaline conditions of those species. These results were consistent with other studies in Mamanguape estuary to investigate spatial patterns of fish assemblages [60, 73]. Similarly, Araújo et al. [74] reported that the mojarra fish species (Gerreidae) used spatial segregation in the use of habitats along the estuarine gradient, while Araújo et al. [75] indicated the environmental variations influencing the distribution of two puffer fish species (*S. testudineus* and *Colomesus psittacus*).

For substrate and primary productivity, these variables could be associated to habitat selection and to foraging preferences showed by species. As observed, a spatial gradient occurred across the estuary: while the upper estuary showed higher values of coarse sand, the lower estuary showed lower values. This result is due to the usual presence of habitats associated to muddy substrate at this location, such as mudflats, seagrass beds, and mangrove which could support a fish diversity that use those habitats differently along ontogenetic development [76, 77]. Substrate type may support differently the benthic macroinvertebrate diversity which is the primary food source for many fish species [78]. Spatial pattern presented by chlorophyll *a* decreased in concentration from upper to lower estuary. The upper estuary is dominated by the frequency of primary producer microphytobenthos observed by Figueiredo and Pessanha [79] and phytoplankton observed by Santana et al. [80], that was favoured due to higher runoff containing freshwater species and nutrients, hence, increasing food availability for the higher level of the food chain. In accordance with Figueiredo and Pessanha [79] and Claudino et al. [81], the upper portion of Mamanguape estuary was dominated by omnivorous and detritivores which increased food web complexity and the lower estuary was dominated by zoobenthivore or carnivore species, such as *C. latus*, *O. saurus*, and *S. greeleyi*. In addition, the abundance of many Engraulidae juveniles and larvae conformed with the higher primary productivity at the upper estuary, probably also due to zooplankton abundance [82]. Most of the Engraulidae species are known by their filter feeding behaviour, for example, species of *Anchoa* sp. [83]. Zooplankton prey have been found in the gut content of small juveniles of several fish species which take advantage of their availability due to abundance and ease of predation [77], since small juvenile fishes have low ability to forage and are vulnerable to predation risk [84].

Moreover, shifts in spatial and seasonal abundance of estuarine fishes could arise as a function of landscape variables, as was demonstrated in our results. For instance, *A. lineatus* and *C. boleosoma* were related to longest distance, while *C. latus* and *A. vulpes* leptocephalus larvae were associated to lower distance from estuary mouth. Site distance from the mouth estuary reflects connectivity degrees with adjacent coastal habitats. As distance decreased, movement of marine fish across sites at the lower estuary could be frequently observed [85, 86]. Such marine species are represented by stenohaline or marine straggle species which occasionally enter the estuary to explore its resources [87]. In addition, spatial connectivity increases the number of rare species in the estuary, which could directly impact functional diversity [3].

The diversity pattern of fish assemblages in this tropical estuary evidenced a change in contribution of α- and β-diversities for γ-diversity from lower to higher scales for both seasons. In this case, the α-component contributed more in sample units than species dissimilarity ($\beta_1$), while scales increased, species dissimilarities ($\beta_2$ and $\beta_3$) have a greater importance for γ-diversity. Thus, this result indicates that changes in estuarine diversity were affected by heterogeneity within the estuarine zone by indirect effects of environmental drivers through beta components. For instance, a higher abundance and richness of larvae and juvenile fishes registered in zone 3 (lower estuary) seemed to be associated primarily to (1) an increase in the influence of marine conditions [10, 55], and/or (2) a close proximity to unstructured (beaches and mudflats) and structured habitats (seagrass beds) found in this zone [76]. Some studies conducted in the Mamanguape estuary have detected evidence of the presence of multiple habitats in the lower part of the estuary that may thus promote an increase in occurrence/abundance of fish species [72, 77]. Furthermore, the presence of multiple habitats is known to increase feeding, sheltering and reproduction opportunities for coastal fish to maximize their probability of survival [12, 87].

Species turnover was also the main component which better explained seasonal changes in beta diversity patterns of fish assemblages. In general, the contributions to variations in dissimilarity among zones ($\beta_3$) was higher in the rainy than dry season. This difference is most observed in the rainy season due to increases in the influx of young-of-the-years in tropical estuaries related to recruitment success [88]. It is not surprising that small juveniles are recruiting into suitable habitat influenced by decreasing salinity and increase of turbidity during the rainy season in tropical estuaries [1, 89]. Such factors are known to have a positive impact on growth rates of fishes, as well as contributing to the increases in food resources and shelter from predators in shallow waters [89–91]. Moreover, during the rainy season juvenile stenohaline species, which tolerate a narrow range of salinity, occurred in higher abundance principally in lower zones related to more stable conditions, and this is likely the result of the proximity of this zone to the ocean and the higher tidal influence. On the other hand, when freshwater inputs are low during the dry season, the marine species move upwards possibly due to the use of the estuary as a nursery associated with a part of their life cycle. In accordance with Dolbeth et al. [92], salinity becomes more stable in the dry season due to the increasing influence of marine water and less influence of freshwater in upper areas in Mamanguape estuary. It is worth highlighting that connectivity with neighbouring coastal habitats is an important feature for recruitment, hence, increased species diversity, as observed by Henderson et al. (2017) [93] to fish assemblages from Moreton Bay, Australia and Henderson et al. [41] to 30 other estuaries across Australia.

In our results, alpha diversity was lower than expected by chance suggesting that species distribution in sample units was random. In other words, it did not have any species sorting since at this level the sites are uniform at this scale. However, as long as spatial scales increased, β-diversity values were higher among sites and zones, being higher than expected by chance pointing to some niche-based process driving species distribution. As suggested by Erös (2007) [94], when higher values of alpha component and higher similarity among sites were observed, lower dissimilarity would be found in upper levels whereas species composition would be the same and habitat homogenization would be observed, hence, decreasing diversity in larger scales. Therefore, our results support the idea that large species dissimilarity in beta components among sites and zones ($\beta_2$ and $\beta_3$) was important to maintaining gamma diversity, while heterogeneity was the factor that set up the sorting species in this estuary. Areas with higher β-diversity likely have larger heterogeneity [95, 96]. Environmental heterogeneity increases dissimilarity in beta diversity because great habitat variability showed different features, thus sustaining distinguished species compositions among sites which are adapted to

different environmental conditions [97]. The increase in dissimilarity at higher levels pointed to differences in functional traits presented by species pools which positioned them in each range of environmental gradient [3].

In conclusion, this study showed how fish assemblages were maintained by environmental heterogeneity, which promoted species replacement (turnover) across gradients acting on assortment and fish movement toward habitats with suitable conditions in both seasons. Additionally, species dissimilarity and gamma diversity were influenced by heterogeneity that allowed change in composition in coarse scale. Therefore, these findings that environmental gradient is important to estuarine fish diversity might be a way by which management and conservation policies are established, and should target turnover maintenance. Despite the estuary being located within the Environmental Protected Area, there are several economic activities present that promote pressure in this ecosystem [49–51, 60]. The ongoing impacts of these activities must be monitored to ensure the maintenance of estuarine habitats, since many fish species depend on several habitats to complete their life cycle [12]. Thus, future conservation policies should aim at maintaining environmental heterogeneity and connectivity to guarantee the nursery ground role to estuarine fish assemblages.

## Supporting information

**S1 Appendix. List of fish species collected in the Mamanguape River estuary, Brazil.** (DOCX)

## Acknowledgments

We thank students of the Laboratory of Fish Ecology (UEPB) for helping in the fieldwork. We would also like to thank the anonymous referees whose suggestions and corrections helped to improve the manuscript.

## Author Contributions

**Conceptualization:** André Luiz Machado Pessanha.

**Formal analysis:** Fernando José König Clark.

**Investigation:** Caroline Stefani da Silva Lima, Emanuelle Bezerra Maciel.

**Methodology:** Caroline Stefani da Silva Lima, Emanuelle Bezerra Maciel, André Luiz Machado Pessanha.

**Supervision:** André Luiz Machado Pessanha.

**Writing – original draft:** Caroline Stefani da Silva Lima, Emanuelle Bezerra Maciel.

**Writing – review & editing:** Fernando José König Clark, André Luiz Machado Pessanha.

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
