## [Decision Letter · Decision Letter 0]

14 Apr 2022

PONE-D-22-05724

Does environmental heterogeneity explain β diversity of estuarine fish assemblages? Example from a tropical estuary under the influence of a semiarid climate, Brazil

PLOS ONE

Dear Dr. Pessanha,

Thank you for submitting your manuscript to PLOS ONE. After careful consideration, we feel that it has merit but does not fully meet PLOS ONE’s publication criteria as it currently stands. Therefore, we invite you to submit a revised version of the manuscript that addresses the points raised during the review process.

The manuscript is of great interest to the scientific community and has publication potential. However, after analyzing the reviewers' comments and after some studies, consider that the authors need to make adjustments before a final opinion.

We look forward to receiving your revised manuscript.

Kind regards,

Luiz Ubiratan Hepp

Academic Editor

PLOS ONE

Journal Requirements:

3. In your Methods section, please include a comment about the state of the animals following this research. Were they released, euthanized or housed for use in further research? If any animals were sacrificed by the authors, please include the method of euthanasia and describe any efforts that were undertaken to reduce animal suffering.

4. Thank you for stating the following financial disclosure: "Unfunded studies"

7. We note that Figure 1 in your submission contain map images which may be copyrighted. All PLOS content is published under the Creative Commons Attribution License (CC BY 4.0), which means that the manuscript, images, and Supporting Information files will be freely available online, and any third party is permitted to access, download, copy, distribute, and use these materials in any way, even commercially, with proper attribution. For these reasons, we cannot publish previously copyrighted maps or satellite images created using proprietary data, such as Google software (Google Maps, Street View, and Earth). For more information, see our copyright guidelines: http://journals.plos.org/plosone/s/licenses-and-copyright.

Additional Editor Comments (if provided):

The proposal is very well grounded for the authors' interest in the spatial scale, however, the introduction is fragile when it presents arguments for the time scale. In addition, the authors do not have a clear time scale, as the collections were carried out in two seasonal periods. This should be better explained in the text.

I believe that the time scale by the aforementioned authors refers to intra-annual (seasonal).

The authors hypothesis that beta diversity by turnover mechanism. And this is explained by the expectation of environmental variability in the reservoirs. Why a hierarchical design? With this design, I hope the authors have some hypothesis for variability at different spatial scales (eg sites, estuaries). Another important point to what? in relation between two periods of the year What is the hypothesis for this? Are there waits for biological occurrences?

The authors cite that environmental heterogeneity will be responsible for the dissimilarity of communities. Therefore, it should be noted that this environmental heterogeneity is related to the spatial scale and the shared component (env+spathy) may be responsible for the greater explanation of the variability of the data. Explore the arguments for this point.

I suggest a description of 'temporal variation' in the study, time scale is not replicated review the study properly. What we can observe are seasonal variables in an annual cycle.

As the authors test the features being land features as the landscape information is estuary scales, and this factor is not replicated. Explain.

All statistical analyzes were performed considering dry and rainy seasons separately, why not the explained variation (env/spatial)?

Review the degrees of freedom for line 330. Are they correct?

I suggest that the authors review the criteria for typing errors and to meet the submission rules.

The study of the study new points not foreseen in the hypotheses of the study. Recommendation to review both topics. I believe the manuscript has more potential than presented.

Reviewers' comments:

Reviewer's Responses to Questions

**Comments to the Author**

1. Is the manuscript technically sound, and do the data support the conclusions?

Reviewer #1: Yes

Reviewer #2: Yes

2. Has the statistical analysis been performed appropriately and rigorously? 

Reviewer #1: Yes

Reviewer #2: Yes

3. Have the authors made all data underlying the findings in their manuscript fully available?

Reviewer #1: Yes

Reviewer #2: Yes

4. Is the manuscript presented in an intelligible fashion and written in standard English?

Reviewer #1: Yes

Reviewer #2: Yes

5. Review Comments to the Author

Reviewer #1: Dear authors,

The review of the manuscript PONE-D-22-05724 is completed. It´s a good contribution for Neotropical ichthyology. I strongly suggest you to include the list of all 125 species as a supplementary file. All other suggestions and comments are in the manuscript file, attached.

Reviewer #2: The manuscript is quite interesting and well written. I send some corrections and suggestions:

1. Bibliographic references must be fully revised according to the journal's formatting rules. This includes citations in the body of the manuscript, which must be presented in numerical order.

2. Some citations were not found in the references: line 50-Datry et al 2017; line 211-The R Development Core Team 2021 (see line 850-R Core Team, 2021); line 92-Anderson 2011; line 261-Baselga et al 2010; line 456-Clark et al 2014; 468-da Silva and Fabré 2019; 491-Garcia and Pessanha 2017; 528-da Silva and Fabré 2019.

3. Some references were not found in the citations: lines 598; 601; 635; 682; 803; 881.

4. The reference 655 is undated.

5. In line 337 wouldn't it be "of abundance in zone 3"?

6. Suggestion 1: when you describe diversity (2.5.3 line 235), you could use: "B1 corresponds to the variation of diversity between the sample subunits", as used in line 357.

7. Suggestion 2: Do not use the terms "semiarid" and "Tropical estuary" as keywords, as they are already included in the title of the manuscript.

6. PLOS authors have the option to publish the peer review history of their article (what does this mean?). If published, this will include your full peer review and any attached files.

Reviewer #1: No

Reviewer #2: No

---

## [Author Response · Author response to Decision Letter 0]

19 Jul 2022

The modifications were accepted and are in the "response to reviewers" file

---

## [Editor Report · Decision Letter 1]

15 Aug 2022

Does environmental heterogeneity explain β diversity of estuarine fish assemblages? Example from a tropical estuary under the influence of a semiarid climate, Brazil

PONE-D-22-05724R1

Dear Dr. Andre Pessanha,

We’re pleased to inform you that your manuscript has been judged scientifically suitable for publication and will be formally accepted for publication once it meets all outstanding technical requirements.

Kind regards,

Luiz Ubiratan Hepp

Academic Editor

PLOS ONE

---

## [Editor Report · Acceptance letter]

2 Sep 2022

PONE-D-22-05724R1 

Does environmental heterogeneity explain β diversity of estuarine fish assemblages? Example from a tropical estuary under the influence of a semiarid climate, Brazil 

Dear Dr. Pessanha:

I'm pleased to inform you that your manuscript has been deemed suitable for publication in PLOS ONE. Congratulations! Your manuscript is now with our production department. 

Kind regards, 

on behalf of

Dr. Luiz Ubiratan Hepp 

Academic Editor

PLOS ONE